# OpenReview forum: "Iteratively Refined Behavior Regularization for Offline Reinforcement Learning"
_NeurIPS.cc/2024/Conference — NeurIPS 2024 poster_

### Official Review · Reviewer_k25f · 2024-07-05

**Soundness:** 2
**Presentation:** 3
**Contribution:** 3
**Rating:** 5
**Confidence:** 4

**Summary:**

The authors introduce a modified version of prior behavior-regularized offline RL methods based on conservative policy iteration, where the current policy is regularized towards an older policy. Performance benefits are demonstrated in the D4RL benchmark.

**Strengths:**

- Easy to implement/add to TD3+BC or other closely related methods.
- Strong performance across multiple offline datasets.
- Significant number of ablations.

**Weaknesses:**

**Skepticism**
- I’m not sure the algorithm does what the authors claim it does. If the current policy escapes the support of the behavior policy (which is possible since the KL to an older policy is a penalty and not a constraint), then the next iteration will not be a refined version of the behavior policy. Instead, like CPI, the update simply penalizes large changes to the policy.

**Experiments**
- Results are based on per-environment hyperparameter optimization. While the authors do compare against a similarly-swept version of TD3+BC, there are other possible explanations for why there are hyperparameter-specific benefits. For example, 2 hyperparameters versus 1 provides more opportunities to overfit. As mentioned below, there is some discrepancy between KL and MSE which may also explain some performance differences.
- Anonymous code link is expired, code is unavailable.
- Minor: The authors used KL for the behavior regularization, but talk about comparisons to TD3+BC, which uses a deterministic policy and minimizes MSE to the behavior policy instead. So I’m not sure how this is implemented and what the TD3+BC baseline uses.

**Questions:**

I imagine an important hyperparameter would be the delay between the current policy and older policy, how much does this impact performance of the algorithm?

**Limitations:**

Satisfactory.

---

> ### Author Rebuttal · Authors · 2024-07-31
>
> ## Whether the algorithm does what the authors claim it does.
> While Proposition 1 establishes that the exact solution of Eq 9 remains within the data support when the actor is initialized as $\pi_\omega=\pi_D$, practical implementations often rely on a limited number of gradient descent steps for optimizing Eq 9, thus could suffer from our-of-support samples. This leads to policy optimization errors  which are further exacerbated iteratively. To approximate the in-support learning, we find it is useful to add the original behavior regularization, which further constrains the policy on the support of data to enhance learning stability. As different datasets have different support, it’s necessary to help the policy stay in the support of the dataset by tuning the hyperparameter across different environments. However, it’s not hard to tune the parameter across different environments and datasets. In Section 5.3.4, we provide ablations of the effect of the two hyperparameter and also summarize some empirical experiences in the section to help users find the suitable hyperparameters when encountering new environments.
>
> ## Results are based on per-environment hyperparameter optimization.
>
> We also provide results of CPI with as little as possible hyper-parameter tuning on the same domain in Table 5 in our appendix. The results show that compared with IQL, CQL and TD3BC, the overall performance of CPI with as little as possible hyper-parameter tuning is still significantly better.
>
> As analyzed before, the main reason to tune the hyper-parameter is to approximate the in-support learning according to the property of different datasets. For example:
>
> - $\lambda$ : When $\lambda=0.1$, the early-stage performance excels, as the behavior policy assists in locating appropriate actions in the dataset. However, this results in suboptimal final convergence performance, attributable to the excessive behavior policy constraint on performance improvement. For larger values, such as 0.9 , the marginal weight of the behavior policy leads to performance increase during training. Unfortunately, the final performance might be poor. This is due to that the policy does not have sufficient behavior cloning guidance, leading to a potential distribution shift during the training process. Consequently, we predominantly select a $\lambda$ value of 0.5 or 0.7 to strike a balance between the reference policy regularization and behavior regularization. In practice, for dataset of higher quality and lower diversity (e.g., expert dataset), we encourage authors to try small $\lambda$. For datasets of lower quality and higher diversity (e.g., medium dataset), larger $\lambda$ should be chosed.
> - $\tau$ : The regularization parameter $\tau$ plays a crucial role in determining the weightage of the joint regularization relative to the Q-value component. We find that (Figure 6 b) $\tau$ assigned to dataset of higher quality and lower diversity (e.g., expert dataset) ought to be larger than those associated with datasets of lower quality and higher diversity (e.g., medium dataset).
>
> ## Anonymous code link is expired
>
> We have modify the link and it could be open now. Please check the anonymous  link in the paper.
>
> ## discrepancy between KL and MSE
>
> Actually $D_{K L}\left(\pi_w(s) \| \bar{\pi}(s)\right)$ was implemented in the experiment  as an MSE loss like $1 / N \sum\left(\pi_w(s)-\bar{\pi}(s)\right)^2$.  For two gaussians, the KL between them can be calculated as $K L(p, q)=\log \frac{\sigma_2}{\sigma_1}+\frac{\sigma_1^2+\left(\mu_1-\mu_2\right)^2}{2 \sigma_2^2}-\frac{1}{2}$. As $\sigma_1$ equals to $\sigma_2$ in the two gaussians, we can obtain the simplified form of this KL as an MSE loss as you described. For deterministic policies, they could be seen as the mean of a gaussian. Therefore, MSE is the specific implementation of KL in our work, so does in TD3BC. That's also the reason we claim that our  algorithm is easy to implement, requiring only a few lines of code modification to existing method, i.e., TD3+BC.  In our provided codes, you could find the corresponding implementation details.
>
>
> ## the delay between the current policy and older policy, how much does this impact performance of the algorithm
>
> The default update interval (UI) of the reference policy in CPI is set to 2 gradient steps. We ablate the update interval 4 and 8 here on six datasets across three seeds in the below table:
>
> |  | CPI  (UI=2) | CPI  (UI=4) | CPI  (UI=8) |
> | --- | --- | --- | --- |
> | halfcheetah-medium | 64.4 $\pm$ 1.6 | 65.3 $\pm$ 0.8 | 61.2 $\pm$ 6.5 |
> | hopper-medium | 98.5 $\pm$ 4.4 | 83.2 $\pm$ 3.6 | 61.8 $\pm$ 44.6 |
> | walker2d-medium | 85.8 $\pm$ 1.0 | 85.6 $\pm$ 0.9 | 81.2 $\pm$ 5.9 |
> | halfcheetah-medium-replay | 54.6 $\pm$ 1.5 | 52.8 $\pm$ 0.4 | 45.4 $\pm$ 14.0 |
> | hopper-medium-replay | 101.7 $\pm$ 1.4 | 89.7 $\pm$ 13.2 | 96.4 $\pm$ 10.2 |
> | walker2d-medium-replay | 91.8 $\pm$ 2.2 | 81.9 $\pm$ 1.7 | 66.3 $\pm$ 35.4 |
>
> It could be seen that increase the update interval of reference policy in CPI could have overall negative impact on the policy’s performance. While with UI=4  on some datasets (halfcheetah-medium, walker2d-medium) it seems there isn’t significant negative influence, in other situations the performance is severely dropped. This may be attributed to the delayed update causes the performance gap between learning policy and the reference policy become larger, that is, the reference policy will remain worse performance for longer time. Thus the reference policy could be more possible to introduce more instability to the training and drag down the performance of the learning policy.

---

> > ### Author Response · Authors · 2024-08-11
> >
> > Thank you for your review and comments. We hope that our additional evaluations and rebuttal have addressed your primary concerns with our paper. We would really appreciate feedback as to whether there are any (existing or new) points we have not covered, and we would be happy to address/discuss them!

---

> > ### Comment · Reviewer_k25f · 2024-08-12
> >
> > Thanks for the response. I have read the rebuttal as well as the other reviews. At this time, I will maintain my original score.
> >
> > Thank you for adding the additional UI experiments. However, 2/4/8 are all in very similar magnitudes, why not values closer to the implicit target network update rate, i.e., 200 or 1000?
> >
> > Concerning the per-environment hyperparameter optimization, little as possible tuning implies that some amount of tuning is required. Note there are baselines that don't require this tuning, and I think that is a serious drawback to this approach.

---

> ### Author Response · Authors · 2024-08-12
>
> Thanks for your reply!
>
> Actually, from the results we can see that  UI=8 has already severely drag down the algorithm's performance. Thus it's highly likely increasing the update rate further could cause more negative influence. In addition, as far as we know, the mentioned large target updating rate are nomally used in the Q-function updating instead of the policy updating. In the standard TD3  algorithm, the target policy's  UI is set to 2. And in the recent SOTA off-policy RL method CrossQ [1], the target policy updating interval is set to 3. Thetefore, from both our empirical results and the setting of other works, it's highly likely the policy updating interval would be better for the performance at a small  value.
>
> It's true that there are baselines that don't require tuning in their paper, such as TD3+BC on MuJoCo domain, which is known for its simplicy. However, the unchanged setting could cause the algorthm perform quite horrible on other domains such as AntMaze and Adroit. See CORL code repository for detailed results (https://github.com/tinkoff-ai/CORL/tree/main).  In addition, several recent SOTA works such as XQL [2], Diffusion-QL [3], STR [4] and SVR [5] on different domains. Therefore, we believe hyperparameters tuning are neccessary for most algorithms on different domains to achieve satisfying performance.  We will also delve into some techniques to automatically optimize the hyperparameters in the future works.
>
>
> We sincerely hope this could address your concerns. We look forward to receiving your further feedback!
>
>
>
> [1] CrossQ: Batch Normalization in Deep Reinforcement Learning for Greater Sample Efficiency and Simplicity. ICLR 2024.
>
> [2] Extreme Q-Learning: MaxEnt RL without Entropy. ICLR 2023.
>
> [3] Diffusion Policies as an Expressive Policy Class for Offline Reinforcement Learning. ICLR 2023.
>
> [4] Supported Trust Region Optimization for Offline Reinforcement Learning. ICML 2023.
>
> [5] Supported Value Regularization for Offline Reinforcement Learning .NeurIPS 2023.

---

### Official Review · Reviewer_h61i · 2024-07-11

**Soundness:** 4
**Presentation:** 4
**Contribution:** 2
**Rating:** 7
**Confidence:** 3

**Summary:**

The authors propose a new offline RL algorithm based on the on conservative policy iteration. The main idea is that the reference policy used for behavior regularization is iteratively modified. The practical algorithm is implemented as a simple modification over TD3-BC, where an additional regularization term is added to control the distance between the current policy being trained, and a frozen version of it. The authors provide some theoretical guarantees in the tabular settings, and evaluate the algorithm on a number of standard benchmarks.

**Strengths:**

## Strengths

1. The paper is well written, and the claims are supported with experiments.

2. The proposed algorithm is simple, and can be implemented with minimal modifications to existing ones

3. The experimental evaluations are extensive

4. Provides theoretical results in the tabular setting

**Weaknesses:**

**1. The algorithm is highly dependent on the value of $\tau$**

The algorithm seems to be dependent on $\tau$, with some experiments requiring a value of 200, while others use values in the range [0.05,2].  Further, the authors choose the value of $\tau$ based on the quality of the dataset, which in most real-world scenario is hard to determine.

**2. BC seems to dominate performance**

Having high values of $\tau$ essentially reduces the offline RL problem to Behavior Cloning. I would encourage the authors to include baselines such as %BC in their results (shown in figure 1), this will help understanding the role $\tau$ plays.

**Questions:**

See Weaknesses.

**Limitations:**

The authors discuss their limitations.

---

> ### Author Rebuttal · Authors · 2024-07-31
>
> ### **The algorithm is highly dependent on the value of $\tau$**
>
> In Section 5.3.4, we provide ablations of the effect of the two hyperparameters. We also summarize some empirical experiences in the section. The regularization parameter $\tau$ plays a crucial role in determining the weightage of the joint regularization relative to the Q-value component. We find that (Figure 6 b) $\tau$ assigned to dataset of higher quality and lower diversity (e.g., expert dataset) ought to be larger than those associated with datasets of lower quality and higher diversity (e.g., medium dataset).
>
> Therefore, in real-world scenario, if you have a impression of the data quality, you could tune the parameters in a confident way. For example, in the recommendation system or the autonomous driving systems, you could directly distinguish and identify  the quality of data collected by the previous deployed algorithms via Return on Investment (ROI) or Success Rate. If the ROI of a policy is much more better than the random policy, one can try to $\tau$ to a relative higher value. If the Success Rate of an autonomous driving car is quite low, then the  $\tau$ should be set to a lower value.
>
> In addition, we also provide results of CPI with as little as possible hyper-parameter tuning on the same domain in Table 5 in our appendix. The results show that compared with IQL, CQL and TD3BC, the overall performance of CPI with as little as possible hyper-parameter tuning is still significantly better.
>
> ### **BC seems to dominate performance**
>
> It’s true that high values of $\tau$ is essential when the dataset is of higher quality (i.e., higher rewards). However, when the dataset is of lower quality (i.e., lower rewards), the Q loss plays more  important role. As you suggested, we include different \%BC (run behavior cloning on only the top X% of timesteps in the dataset, ordered by episode returns) in our results, citing from the Decision Transformer paper:
>
> |  |  10%BC   | 25%BC  | 40%BC | 100%BC | CPI |
> | --- | --- | --- | --- | --- | --- |
> | halfcheetah-medium | 42.9  | 43.0  | 43.1 | 43.1 | **64.4** |
> | hopper-medium | 65.9  | 65.2  | 65.3  | 63.9 | **98.5** |
> | walker2d-medium |  78.8  | 80.9  | 78.8  | 77.3 | **85.8** |
> | halfcheetah-medium-replay | 40.8  | 40.9  | 41.1   | 4.3 | **54.6** |
> | hopper-medium-replay | 70.6  | 58.6 |  31.0  | 27.6 | **101.7** |
> | walker2d-medium-replay | 70.4  | 67.8 |  67.2  | 36.9 | **91.8** |
>
> It can be observed on these datasets of lower quality, \%BC indeed help improve over the original BC (100\%BC). However, they still fall behind CPI.  We believe this could help you understand $\tau$'s important role in CPI’s performance.

---

> > ### Comment · Reviewer_h61i · 2024-08-07
> >
> > I thank the authors for their response, I have updated my score accordingly, and wish the authors the best!

---

### Official Review · Reviewer_Hcq2 · 2024-07-12

**Soundness:** 3
**Presentation:** 3
**Contribution:** 3
**Rating:** 7
**Confidence:** 4

**Summary:**

The paper introduces Conservative Policy Iteration (CPI), a new policy regularization algorithm for offline reinforcement learning. The core concept behind this approach is the iterative refinement of the reference policy for regularization. The algorithm guarantees policy improvement while avoiding out-of-sample actions and converges to the in-sample optimal policy. The paper also discusses practical implementations of CPI for continuous control tasks and evaluates its performance on the offline RL benchmarks.

**Strengths:**

* The idea of iteratively refining the reference policy provides a new approach to address the challenges of typical behavior regularization. This contribution improves the robustness and performance of behavior regularization methods.
* The paper provides theoretical analysis to support CPI in the tabular setting and demonstrates its superior performance compared to previous methods in empirical evaluations. The experiments are well-designed and provide comprehensive results. The theoretical analysis is also presented in a clear and concise manner.
* The presentation is good. The authors provide clear explanations of the algorithm, its implementation details, and the experimental setup.

**Weaknesses:**

* The authors have mentioned that one of the limitations of CPI is the need for the selection of two hyperparameters. It would be helpful to have a detailed evaluation and discussion on this. For example, ablation on additional offline datasets would offer a deeper insight.
* The detailed experimental setup of Figure 1 is missing.

**Questions:**

* Can you provide more detailed hyperparameter study results, including those on additional datasets?
* Are there any specific types of tasks or environments where CPI may underperform typical behavior regularization method TD3+BC under the same regularization strength?

**Limitations:**

The authors have stated the limitations of this work.

---

> ### Author Rebuttal · Authors · 2024-07-31
>
> ## more detailed discussion and hyperparameter study results
>
> In Section 5.3.4, we provide ablations of the effect of the two hyperparameters. We also summarize some empirical experiences in the section.
>
> - $\lambda$ : When $\lambda=0.1$, the early-stage performance excels, as the behavior policy assists in locating appropriate actions in the dataset. However, this results in suboptimal final convergence performance, attributable to the excessive behavior policy constraint on performance improvement. For larger values, such as 0.9 , the marginal weight of the behavior policy leads to performance increase during training. Unfortunately, the final performance might be poor. This is due to that the policy does not have sufficient behavior cloning guidance, leading to a potential distribution shift during the training process. Consequently, we predominantly select a $\lambda$ value of 0.5 or 0.7 to strike a balance between the reference policy regularization and behavior regularization. In practice, for dataset of higher quality and lower diversity (e.g., expert dataset), we encourage authors to try small $\lambda$. For datasets of lower quality and higher diversity (e.g., medium dataset), larger $\lambda$ should be chosed.
> - $\tau$ : The regularization parameter $\tau$ plays a crucial role in determining the weightage of the joint regularization relative to the Q-value component. We find that (Figure 6 b) $\tau$ assigned to dataset of higher quality and lower diversity (e.g., expert dataset) ought to be larger than those associated with datasets of lower quality and higher diversity (e.g., medium dataset).
>
> In addition, we also provide results of CPI with as little as possible hyper-parameter tuning on the same domain in Table 5 in our appendix. The results show that compared with IQL, CQL and TD3BC, the overall performance of CPI with as little as possible hyper-parameter tuning is still significantly better.
>
> ## ablation on additional offline datasets
>
> In the PDF of the general response, we provide more hyperparamerts ablations. All the results are averaged across three seeds using the final obtained model after training for 1M gradient steps. The results further prove our empirical experiences of hyperparameter tuning provided above.
>
> ## detailed experimental setup of Figure 1
>
> In Figure 1 we empirically demonstrate the impact of regularization utilizing distinct policies on the 'hopper-medium-replay' and 'hopper-medium-expert' datasets in D4RL. We leverage Percentile Behavior Cloning (\%BC) to generate policies of varied performance \citep{chen2021decision}. Specifically, for behavioral cloning, we filter trajectories of the top 5\%, median 5\%, and bottom 5\% returns. Following this, we modify TD3+BC to develop the TD3+5\%BC algorithm by replacing the behavior regularization with the 5\%BC policy, and subsequently train TD3+5\%BC on the original dataset. These descriptions are also provided in the introduction section. We’ll try to make them clearer to readers.
>
> ## tasks or environments where CPI may underperform typical behavior regularization method TD3+BC under the same regularization strength
>
> In Table 8 in our appendix, we ablate different variants of TD3+BC and compare CPI with them. TD3+BC is set $\alpha$ to a constant value of 2.5 for each dataset, whereas CPI chooses the appropriate $\tau$ from a set of $\tau$ alternatives. We note that the hyperparameter that plays a role in regulating Q and regularization in CPI is $\tau$, which can essentially be understood as the reciprocal of $\alpha$ in TD3+BC. Therefore, for the convenience of comparison, we rationalize the reciprocal of $\tau$ as the parameter $\alpha$. In this section, we set the $\alpha$ of TD3+BC to be consistent with that of CPI in order to show that the performance improvement of CPI mainly comes from amalgamating the benefits of both behavior-regularized and in-sample algorithms. Further, we also compare CPI with TD3+BC with dynamically changed $\alpha$ and TD3+BC with swept best $\alpha$ in the ranges {0.0001, 0.05, 0.25, 2.5, 25, 36, 50, 100}, which improves TD3+BC by a large margin, to show the superiority of CPI. The selection of parameters is shown in Table 4.
>
> The results for TD3+BC (vanilla), TD3+BC (same $\alpha$ with CPI), TD3+BC (swept best $\alpha$), TD3+BC with dynamically changed $\alpha$ and CPI are shown in Table 8. Comparing the variants of TD3+BC with different $\alpha$ choices, it can be found that changing $\alpha$ can indeed improve the performance of TD3+BC. However, compared with TD3+BC (same $\alpha$) and TD3+BC (swept best $\alpha$), the performance of CPI is significantly better, which proves the effectiveness of the mechanism for iterative refinement of policy for behavior regularization in CPI.

---

> > ### Comment · Reviewer_Hcq2 · 2024-08-08
> >
> > I thank the authors for their detailed response and for conducting extensive experiments during the rebuttal. I believe they are very helpful and would appreciate their inclusion in the final manuscript. I apologize for having overlooked some contents in the appendix. My concerns have been adequately resolved, and I maintain my positive evaluation of this work.

---

### Official Review · Reviewer_Y6m9 · 2024-07-13

**Soundness:** 2
**Presentation:** 3
**Contribution:** 2
**Rating:** 5
**Confidence:** 4

**Summary:**

Policy constraint is a standard approach to offline RL. Research in this area often involves using different types of divergence to regulate the distance between the current (learned) policy and the behavior (reference) policy. This paper proposes a new perspective on policy constraint offline RL: why not update the reference policy?

By doing so, policy constraint offline RL methods can handle heterogeneous datasets, where trajectories are collected by different levels of behavior policies. This is an interesting, promising, and novel idea. The example in Figure 1 clearly verifies this motivation.

As for the empirical evaluation part, many state-of-the-art baseline algorithms such as IQL, EDAC, and STR are included, covering policy constraint methods and value regularization methods. Empirical comparison on the D4RL benchmark suggests that iteratively refining the reference policy significantly improves performance, especially with the help of a non-algorithmic technique, Reference Ensembles (CPI-RE).

**Strengths:**

1. The idea of iteratively updating the reference policy for policy constraint offline RL is well-motivated.
2. The motivation, implementation, and convergence analysis (towards the unique fixed solution with the in-sample version of tabular MDP) are clear to me.
3. Empirical evaluation includes comprehenseive baseline algorithms.

**Weaknesses:**

The writing can be improved, though it is not technical nor affects the contribution of this paper.
    1. “behind” → “behinds”, Line 57
    2. Proposition 1 has a restatement in the appendix, which has a different number (Proposition 2). I believe the LaTeX command`\begin{restatable}{theorem}` would help. Besides, I think it would be better to replace "optimal policy" with "solution".

**Questions:**

I have a question regarding Proposition 1. I checked the proof in the appendix, where I find that the proof of $E_{a\sim \pi^*}[Q^\pi(s,a)] \geq E_{a\sim \pi}[Q^\pi(s,a)]$ is correct. However, this is different from the conclusion that $V^{\pi^*} \geq V^{\pi}$, because the notation $V^{\pi^*}(s)$ normally refers to the expected returns by following $\pi^*$, that is $E_{a\sim \pi^*}[Q^{\pi^*}(s,a)]$.

**Limitations:**

please see questions.

---

> ### Author Rebuttal · Authors · 2024-07-31
>
> ## The writing can be improved
>
> We sincerely appreciate your suggestions for improving the readability of our paper. We have modified our paper as your suggestion!
>
> ## question regarding Proposition 1.
> You’re right that $E_{a \\sim \\pi^*}\\left[Q^\\pi(s, a)\\right] \\geq E_{a \\sim \\pi}\\left[Q^\\pi(s, a)\\right]$ is different from the conclusion that $V^{\\pi^*} \\geq V^\\pi$. However, $V^{\\pi^*} \\geq V^\\pi$ could be proved in a quite simple way using the Policy Improvement Theorem and the corresponding proof.
>
> **Theorem:** Consider two policy $\\pi(a \\mid s), \\pi^{\\prime}(a \\mid s)$, and define $$ Q^\\pi\\left(s, \\pi^{\\prime}\\right)=\\mathbb{E}_{a \\sim \\pi^{\\prime}(a \\mid s)}\\left[Q^\\pi(s, a)\\right] . $$
>
> If $\\forall s \\in S$, we have that $Q^\\pi\\left(s, \\pi^{\\prime}\\right) \\geq V^\\pi(s)$, then it holds that $V^{\\pi^{\\prime}}(s) \\geq V^\\pi(s), \\forall s \\in S$. This means that $\\pi^{\\prime}$ is atleast as good a policy as $\\pi$.
>
> **Proof**. Note that $Q^\\pi\\left(s, \\pi^{\\prime}\\right) \\geq V^\\pi(s)$. By expanding $Q^\pi$, we can get that $\\forall s \\in S$,
>
> \begin{aligned}
> & V^\pi(s) \leq Q^\pi\left(s, \pi^{\prime}\right) \\\\
> & =\mathbb{E}_{a \sim \pi^{\prime}(a \mid s), s^{\prime} \sim \mathcal{T}^{\prime}\left(s^{\prime} \mid s, a\right)}\left[r(s, a)+\gamma V^\pi\left(s^{\prime}\right)\right] \\\\
> & \\leq \\mathbb{E}\_{a \\sim \\pi^{\\prime}(a \\mid s), s^{\\prime} \\sim \\mathcal{T}^{\\prime}\\left(s^{\\prime} \\mid s, a\\right)}\\left[r(s, a)+\\gamma Q^\\pi\\left(s^{\prime}, \\pi^{\\prime}\\right)\\right] \\\\
> & =\\mathbb{E}\_{a, a^{\\prime} \\sim \\pi^{\\prime}}\\left[r(s, a)+\\gamma r\\left(s^{\\prime}, a^{\\prime}\\right)+\\gamma^2 V^\\pi\\left(s^{\\prime \\prime}\\right)\\right] \\\\
> & \\leq \\ldots \\\\
> & \\leq \\mathbb{E}\_{a, a^{\\prime}, a^{\\prime \\prime} \\ldots \\sim \\pi^{\\prime}}\\left[r(s, a)+\\gamma r\\left(s^{\\prime}, a^{\\prime}\\right)+\\gamma^2 r\\left(s^{\\prime \\prime}, a^{\\prime \\prime}\\right)+\\ldots\\right] \\\\
> & =V^{\\pi^{\\prime}}(s) \\\\
> \end{aligned}
>
>
> This completes the proof that the new policy $\\pi'$ is at least as good as the original policy $\\pi$ in terms of the state-value function for all states $s$.  According to this Policy Improvement Theorem and the proof of Proposition 2 in our paper, we can directly obtain that $V^{\pi^*}(s)=E_{a \sim \pi^*}[Q^{\pi^*}(s, a)] \geq E_{a \sim \pi^*}[Q^\pi(s, a)] \geq E_{a \sim \pi}[Q^\pi(s, a)]= V^\pi(s)$.
>
> We sincerely hope this could address your concerns. If you feel that the manuscript meets the criteria for a higher rating, I would be immensely appreciative of any positive adjustments to the score. Your recognition of the merits of this research would encourage us a lot. I look forward to receiving your feedback and am open to any suggestions that may further enhance the quality of this manuscript.

---

> > ### Comment · Reviewer_Y6m9 · 2024-08-13
> >
> > Thank you for addressing my concerns and revising the proof. After reviewing the updated proof, I agree with the changes. Based on this, I will be increasing my score for your paper, as the revised content strengthens your contribution.

---

> > > ### Author Response · Authors · 2024-08-13
> > >
> > > Thanks for you reply! We'll add the revised  proof in our paper as your suggestion.

---

> ### Author Response · Authors · 2024-08-11
>
> Thank you for your review and comments. We hope that our additional discussion and rebuttal have addressed your primary concerns with our paper. We would really appreciate feedback as to whether there are any (existing or new) points we have not covered, and we would be happy to address/discuss them!

---

> > ### Author Response · Authors · 2024-08-12
> >
> > Dear Reviewer Y6m9,
> >
> > We hope that you've had a chance to read our responses and clarification. As the end of the discussion period is approaching, we would greatly appreciate it if you could confirm that our updates have addressed your concerns.

---

### Author Rebuttal · Authors · 2024-08-01

In the PDF we provide more hyperparamerts ablations.

---

### Decision · Program_Chairs · 2024-09-25

**Decision:**

Accept (poster)

**Comment:**

The reviewers are divided into two camps: enthusiasts and skeptics. As the Area Chair, I align with the former group. I commend the clear exposition of the approach, the thoroughness and rigor of the empirical study, and the impressive performance demonstrated. Based on these strengths, I recommend accepting this submission as a poster. However, I also acknowledge the validity of Reviewer k25f's concerns regarding per-environment hyperparameter optimization. Addressing this issue more thoroughly in the final version could significantly enhance the paper's overall impact.